# Prevalence of violence and associated factors among youth in Northwest Ethiopia: Community-based cross-sectional study

Alehegn Bishaw Geremew[1‡], Abebaw Addis Gelagay[1‡], Telake Azale Bisetegn[2☉], Yohannes Ayanaw Habitu[1☉], Solomon Mekonen Abebe[3☉], Eshetie Melese Birru[4☉], Temiro Azanaw Mengistu[5☉], Yilikal Tiruneh Ayele[6☉], Hedija Yenus Yeshita[1☉]*

1 Department of Reproductive Health, Institute of Public Health, College of Medicine and Health Science, University of Gondar, Gondar, Ethiopia, 2 Department of Health Education and Behavioural Science, Institute of Public Health, College of Medicine and Health Science, University of Gondar, Gondar, Ethiopia, 3 Department of Human Nutrition, Institute of Public Health, College of Medicine and Health Science, University of Gondar, Gondar, Ethiopia, 4 Department of Pharmacology, School of Pharmacy, College of Medicine and Health Science, University of Gondar, Gondar, Ethiopia, 5 Health Department, Central Gondar Zone, Amhara Regional State, Ethiopia, 6 Amhara Regional Health Bureau, Amhara Regional State, Ethiopia

☉ These authors contributed equally to this work.
‡ ABG and AAG also contributed equally to this work.
* kedijayenus@gmail.com

**Data Availability Statement:** All relevant data are within the paper and its Supporting information files.

## Abstract

### Background

Violence as a known serious public health problem affects people in all stages of life, from childhood to the elderly. In society, one of the most visible forms of violence is young people violence, whereas they, adolescents and young adults, are the main victims of such violence. There was limited information on the burden of violence and factors among this age group. Therefore, this study aimed to determine the prevalence of violence and its associated factors among youth in Northwest, Ethiopia.

### Methods

A community-based cross-sectional study design was conducted to estimate the magnitude of violence among youth in Northwest, Ethiopia. Data were taken from the mega project entitled assessment of common health problem and risky health behavior among youth. Youth violence was the dependent variable whereas the socio-demographic variables and substance use were the independent-variables. The bivariate logistic regression model was employed to identify associated factors. An adjusted odds ratio with a 95% confidence interval was used to determine factors associated with violence.

### Results

From the total 1765 representatives of the youth population,1597 (90.5%) youth participated in the study. Overall, the prevalence of violence among youth aged 15–24 years for the last 12 months was 21.5%. In the multivariable logistic regression model, factors significantly associated with violence were being married and divorced (AOR = 1.77, 95%CI: 1.13, 2.79)

**Funding:** a) The University of Gondar supported the staffs financially to conduct this research. b) The funder had no role in this research: had no role in study design, data collection and analysis, decision to publish, or preparation of the manuscript. c) All authors are received their salaries from University of Gondar except Temiro Aza.

**Competing interests:** The authors have declared that no competing interests exist.

**Abbreviations:** AOR, Adjacent Odds Ratio; CI, Confidence Interval; COR, Crude Odds Ratio; DHS, Demographic and Health Survey; EDHS, Ethiopia Demographic and Health Survey; SD, Standard Deviation; UoG, University of Gondar; USA, United States of America; WHO, World Health Organization.

and (AOR = 5.67, 95%CI: 2.93, 10.99), respectively, living with mother's only (AOR = 1.85,95%CI: 1.28, 2.66) and father's only (AOR = 2.45, 95% CI: 1.30, 4.63), and substance use (AOR = 2.38,95% CI: 1.56, 3.66).

## Conclusions

The prevalence of violence among youth was high compared to other studies. Special emphasis Should be given for youth violence in order to manage the victims as well as for preventing and controlling the identified factors through strengthening policies and strategies.

## Background

The World report on violence defines violence as, "the intentional use of physical force or power, threatened or actual, against another person or a group or community that either result in or has a high likelihood of resulting in injury death, psychological harm, mal- development or deprivation" [1, 2]. Violence as a known serious public health problem affects people in all stages of life [3]. In society, one of the most visible forms of violence is young people violence, whereas they, adolescents and young adults, are the main victims of such violence [4]. It is estimated that each year, 200,000 homicides occur in this age group in the world [2]. Nearly all of these deaths occur in low and middle-income countries and the majority of victims (83%) are males [2]. Violence is among the top five leading causes of death for young people [5] and the elimination of violence is foreseen in the United Nations Agenda Sustainable Development Goals by 2030 [3, 6]. Youth violence includes physical, psychological and sexual abuse, neglect, commercial or other exploitation of children (e.g labor exploitation, forced marriage, forced criminality, domestic servitude, child soldiers) "resulting in actual or potential harm to the child' health, survival, development or dignity in the context of establishing a relationship, responsibility, trust or power" [7, 8].

Youth violence often occurs alongside other types of violence. For instance, maltreated children are themselves at increased risk in later life of either perpetrating or becoming the victims of multiple types of violence–including suicide, sexual violence, youth violence, and intimate partner violence. The same set of factors–such as harmful levels of alcohol use, family isolation and social exclusion, high unemployment, and economic inequalities–have been shown to underlie different types of violence [9]. Victims are not the only heirs of youth violence, but also deeply harm their families, friends, and communities. A great increase in the costs of health and welfare criminal justice services decreases the value of property in areas where it occurs, disrupts a range of essential services, reduces productivity, and generally undermines the fabric of society are imposed consequences of youth violence [2].

A study done in Serbia violence among youth was 13.4% [10], and China 13.2% [11]. The prevalence of violence in EDHS report was 27% [12]. A systematic review done in low and middle-income regions reported that, the lowest physical fighting between twenty studied countries was for females in Myanmar (8%), whereas, the highest rates are reported for boys in Samoa (73%) [2].

In Ethiopia, nearly 16% of male college students reported physically abusing [13]. Worldwide, around 15 million adolescent girls aged 15 to 19 have experienced forced sex in their lifetime; 9 million of these girls were victimized within the past year [14]. The reported prevalence of sexual violence among young people in dating relationships varies from 1.2%–32.9% for

females in North America and Europe [15]. According to demographic and health survey data for selected low and middle-income countries, the percentage of girls aged 15–19 years who have ever experienced forced sexual intercourse ranges from zero among adolescent girls in Kyrgyzstan, to 22% among girls in the same age range in Cameroon. Among girls and women aged 15–49 years, the percentage reporting forced sexual initiation ranges from 1% in Timor-Leste to 29% in Nepal [16], in Uganda, 35% [17] and EDHS report, seven percent of women age15-49 reported that they have experienced sexual violence in the past 12 months, five percent of women had experienced sexual violence by age 18, including 2% who had experienced sexual violence by age 15 [12].

A study in Serbia showed that (in the family and on the street) 2.8% and 5.3% were victims of psychological violence, respectively [10]. The experience of any emotional violence in among ever-married women age 15–49 years was 20.2% [12].

The associations of socio-demographic and other factors with violence; being raised in poverty has been found to contribute a greater likelihood of involvement in violence, and poverty both in the community and at the level of individual households has been shown to predict violence [2, 18]. A study conducted in Serbia, predictors of violence victimization were male gender, lack of close friends and urban settlements [10]. The youngest women (age15-19), women with no children, and never-married women are less likely to have experienced violence [12]. Young men are at far greater risk than females for becoming perpetrators and victims of youth violence [19]. Those who are involved in youth violence show lower educational performance and are more at risk of school dropout [20]. At the individual level, young people who start drinking early and drink frequently are at increased risk of perpetrating or being a victim of youth violence [21]. DHS in Ethiopia, all forms of spousal violence are higher among divorced/separated/widowed women, women with no education, with the level of husband's/partners alcohol consumption and urban-rural residence [12].

There are no or limited studies conducted to assess violence against young people and its determinants in the study area as well as in Ethiopia. Even some studies conducted previously focused only on the reproductive health problems of young people; yet, which does not address violence against young people in both sex groups. Hence, the present study was designed to determine the prevalence of violence and related factors among youth in Northwest Ethiopia.

Thus, the findings of this study contribute to public health researchers and policy makers to promote further research and develop or modify strategies designed to reduce violence.

## Materials and methods

### Study design, setting and period

A community-based cross-sectional study design was conducted to estimate the magnitude of violence and associated factors among young people aged 15–24 years. This study was done in Central, West and North Gondar Zones. A total of 24 woredas and one city Administration present in the three zones: Central Gondar Zone have 13 woredas, North Gondar Zone 7 woredas, and West Gondar Zone 4 woredas. The total number of kebeles in the three zones is 546 having a total population of 3,654,920 populations, from this 1,847,631 was males and 1,807,289 females [22]. The study was done from September 2018 to June 2020.

### Sample size and sampling techniques

The source population of the study was young people aged 15–24 years living both in rural and urban settings in Central, West and North Gondar Zone whereas youth in the selected kebeles of in these rural and urban settings were the study population. The sample size of the study was calculated by considering the prevalence of a study conducted in Ethiopia 27.0% [12],

significance level = 95%, margin of error = 3%, and design effect = 2. The total sample size was 1681. By considering the non-response rate of 5%, the final sample size was 1765.

A multistage cluster sampling method was employed to get the study participants.

This study was done in Central, West and North Gondar Zone. A total 24 woredas present in the three zones: Central Gondar Zone have 13 woredas, North Gondar Zone 7 woredas and West Gondar Zone 4 woredas.

Firstly, two woredas from the Central Gondar Zone, one woreda from each North Gondar Zone and West Gondar Zone were selected by simple random sampling. Secondly, three kebeles were selected from the selected woredas. From each kebele, three clusters (Ketena/ Gote) were randomly selected. The non-proportional or equal location was considered to get households in each cluster and eligible study participants were interviewed.

## Study variables

Youth violence is defined by the World Health Organization "the intentional use of physical force or power, threatened or actual, against youth, another person, or against a group or community, that either result in or has a high likelihood of resulting in injury, death, psychological harm, mal-development or deprivation" Any of the specified acts of physical, sexual, or emotional violence in the past 12 months preceding the survey [1, 12].

Emotional violence is say or does something to humiliate you in front of others; threaten to hurt or harm you or someone close to you; insult you or make you feel bad about yourself [12]. Physical violence is push you, shake you, or throw something at you; slap you; twist your arm or pull your hair; punch you with his/her fist or with something that could hurt you; kick you, drag you, or beat you up; try to choke you or burn you on purpose; or threaten or attack you with a knife, gun, or any other weapon [12].

Sexual violence is defined as physical force you to have sexual intercourse with him even when you did not want to; physically force you to perform any other sexual acts you did not want to; force you with threats or in any other way to perform sexual acts you did not want to [12].

The survey questions to address the above variables: Have you experienced violence in the past 12 months? Have you experienced violence like hit /slapped or thrown something by someone in the past 12 months? Have you experienced violence like scared/intimidated by somebody in the past 12 months? Have you exposed to sexual violence for the last 12 months?

## Data collection procedure

Data were collected from the mega project entitled assessments of common health problem and risky health behaviour among youth aged 15–24 years in Central, North and West Gondar Zone. Twenty-four data collectors, who were nurses and midwives working in the public health institutions in the selected woredas especially those who were working in a youth-friendly clinic, and six supervisors (one supervisor per woreda) who were at least degree holders involved during data collection. To assure data quality, training was given for data collectors and supervisors on the objective of the study, consent and confidentiality, and data collection technique. After getting the consent, the data were collected from study participants through pre- tested interviewer-administered questionnaire, and both supervisors and investigators were checked for completeness before data entry.

## Data processing and analysis

An Epi Info version 7 was used for data entry and transferred to STATA 14 for data management and final analysis. First, descriptive analysis was done to determine the problem of

violence among young people. Second, binary logistic regression was used to do both bivariable and multivariable analyses to see the association between covariates and violence. Odds ratio with 95% CI and P-value <0.05 was used to determine the presence and the strength of the association between dependent and independent variables.

### Ethics approval and consent to participate

Ethical clearance was obtained from the Institutional Review Board of University of Gondar. Permission letter to conduct the study was obtained from Central, West and North Gondar Zone Health department. Assent and verbal informed consent were obtained from potential study participants. Verbal consent for age 18 and above, and verbal assent for minors was obtained from their families or guardians after detailed explanation on the purpose, risks, and benefits of the study including the selection of the study participants is randomly and they have the right not to participate or withdraw from the study at any time. To ensure confidentiality, their name and other personal identification were not registered in the format. Finally, the participation of the study participants was confirmed by receiving their responses of agreement and have made tick to each consent form by the data collectors.

## Results

### Socio-demographic characteristics of Youth in Northwest, Ethiopia

From the total 1765 representative sample of the youth population, 1597 (90.5%) young people participated in the study. From the total respondents, 50.7% was found in Central Gondar, 68.1% of the participant's age in the study was below 20 years, the mean and median age of the participants were 19.19 and 19 respectively, with an SD± 2.85. More than half of the respondents 51.2% were females. The majority of respondents was in rural settlements 53.5% and had primary education 46.6%. The majority of their parents (mother and father) were (88.6%) and (80.2%) alive, respectively. The educational status of their parent's not educated, mother's (77.5%) and father's (37.7%) (Table 1).

### The prevalence of violence among youth 15 to 24 years old in the past 12months

Overall, the prevalence of violence among youth for the last 12 months was 21.5%. Youth hit /slapped or thrown something (physical violence), scared/intimidated (psychological violence), and exposed to sexual violence for the last 12 months were 8.3%, 13.6%, and 3.4%, respectively (Table 2).

### Factors associated with youth violence

In the bivariate logistic regression analysis, a statistically significant association (p<0.05) was observed between youth violence and the independent variables. All independent variables listed in the socio-demographic table and other potential variable from the survey (substance use) were tested in the bi-variable analysis. Those variables sex, marital status, educational level, currently attending with school, living arrangement, wealth index, and substance use have a p-value of < 0.2 were taken in to the multivariable logistic regression' model.

The odds of being married and 2.11 divorced (AOR = 1.77, 95%CI:1.13, 2 79) and (AOR = 5.67, 95%CI:2.93, 10.99), respectively associated with violence among youth compared to unmarried or single. Living with mother only (AOR = 1.85, 95%CI: 1.28, 2.66), father's only (AOR = 2.45, 95%CI: 1.30, 4.63) and husband/wife (AOR = 0.44, 95%CI:0.25, 0.76) were associated with violence compared to living with both parents.

**Table 1. Socio-demographic characteristics of youth and youth parent's Northwest Ethiopia.**

| Variable name | Frequency | Percentage |
|---|---|---|
| **Zone** | | |
| Central Gondar | 810 | 50.7 |
| North | 523 | 32.8 |
| West Gondar | 264 | 16.5 |
| **Age** | | |
| <20 | 1087 | 68.1 |
| 20 and above | 510 | 31.9 |
| **Sex** | | |
| Male | 780 | 48.8 |
| Female | 817 | 51.2 |
| **Residence** | | |
| Rural | 854 | 53.5 |
| Urban | 743 | 46.5 |
| **Religion** | | |
| Orthodox | 1483 | 92.9 |
| Muslim | 104 | 6.5 |
| Others | 10 | 0.6 |
| **Marital status** | | |
| Single | 1721 | 79.6 |
| Married | 283 | 17.7 |
| Divorced | 43 | 2.7 |
| **Educational level** | | |
| Unable to read and write | 114 | 7.1 |
| Primary education | 739 | 46.3 |
| Secondary education | 550 | 34.4 |
| Higher | 194 | 12.2 |
| **Currently attending school** | | |
| No | 611 | 38.3 |
| Yes | 986 | 61.7 |
| **Occupation** | | |
| Unemployed | 1356 | 84.9 |
| Employed | 241 | 15.1 |
| **Living arrangement** | | |
| Mother and father | 758 | 47.5 |
| Mother only | 184 | 11.5 |
| Father only | 45 | 2.8 |
| Husband/wife | 222 | 13.9 |
| Relative/sister or brother | 105 | 6.6 |
| Others | 70 | 4.4 |
| Alone | 213 | 13.3 |
| **Mother alive** | | |
| No | 182 | 11.4 |
| Yes | 1415 | 88.6 |
| **Mothers education** | 1415 | |
| Cannot read and write | 997 | 70.8 |
| Read and write | 223 | 15.8 |
| Primary education | 116 | 8.2 |

*(Continued)*

**Table 1.** (Continued)

| Variable name | Frequency | Percentage |
|---|---|---|
| Secondary education | 53 | 3.7 |
| College and above | 26 | 1.8 |
| **Mothers occupation** | | |
| Housewife | 1237 | 77.5 |
| Merchant | 94 | 5.9 |
| Government employee | 35 | 2.2 |
| Private employee | 9 | 0.6 |
| Daily laborer | 26 | 1.6 |
| Others | 14 | 0.9 |
| **Father alive** | | |
| No | 316 | 19.8 |
| Yes | 1281 | 80.2 |
| **Fathers education** | | |
| Cannot read and write | 483 | 37.7 |
| Primary education | 657 | 51.3 |
| Secondary education | 99 | 7.7 |
| Higher education | 41 | 3.2 |
| **Father's occupation** | | |
| Merchant | 245 | 15.3 |
| Government employee | 98 | 6.1 |
| Private employee | 35 | 2.2 |
| Daily laborer | 30 | 1.9 |
| Farmer | 858 | 53.7 |
| Others | 15 | 0.9 |
| **Number of families** | | |
| <5 | 1018 | 63.7 |
| 5–9 | 557 | 34.9 |
| 10 and above | 22 | 1.4 |
| **Wealth index** | | |
| First(lowest) | 319 | 20 |
| Second | 320 | 20 |
| Third | 319 | 20 |
| Fourth | 320 | 20 |
| Highest | 319 | 20 |

In this study substance use is 2.2 times more likely associated with youth violence compared to non-substance use (AOR = 2.38, 95% CI: 1.56, 3.66) (Table 3).

## Discussion

Violence is the leading cause of morbidity as well as mortality among young or youth populations. In this study, the prevalence of violence among youth was 21.5%. This finding was in line with the study conducted in Ethiopia, 27% [12], but it was higher than a study conducted in Serbia 13.4% [10], China 13.2% [11] and in Ethiopia 16% [13]. The higher finding in our study might be most of the youth were unemployed and living in lower socioeconomic status that imposed them on substance use, commits crime and violence.

Table 2. Percentage of youth violence and their personal behavior Northwest, Ethiopia.

| Variable | Frequency | Percentage |
|---|---|---|
| **One people scare or intimidate you for the last 12 months** | | |
| No | 1380 | 86.4 |
| Yes | 217 | 13.6 |
| **Intimidate for the last 3 months** | | |
| No | 86 | 39.6 |
| Yes | 131 | 60.4 |
| **Threatened to hurt you for the last 12 months** | | |
| No | 1471 | 92.1 |
| Yes | 126 | 7.9 |
| **Threatened to hurt you for the last 3 months** | | |
| No | 40 | 31.7 |
| Yes | 86 | 68.3 |
| **Hit/slapped you or thrown something to you for the last12 months/physical violence** | | |
| No | 1465 | 91.7 |
| Yes | 132 | 8.3 |
| **Hit/slapped you or thrown something to you the last 3months** | | |
| No | 43 | 32.6 |
| Yes | 89 | 67.4 |
| **Forced or pressured you to have sexual intercourse for the last 12 months/sexual violence** | | |
| No | 1543 | 96.6 |
| Yes | 54 | 3.4 |
| **Forced or pressured you to have sexual intercourse for the last 3 months** | | |
| No | 32 | 59.3 |
| Yes | 22 | 40.7 |
| **Violence within the last 12 months** | | |
| No | 1254 | 78.5 |
| Yes | 343 | 21.5 |
| **Violence within the last 3months** | | |
| No | 1371 | 85.8 |
| Yes | 226 | 14.2 |
| **Ever use substance** | | |
| No | 1489 | 93.2 |
| Yes | 108 | 6.8 |

In this study, the prevalence of physical violence among youth in the past 12 months was 8.3%. It was comparable with a study done in Serbia7.3% [10], Myanmar (8%) [2], and in Malaysia, 11.8% [23], but lower than a review done in twenty countries (in low and middle-income regions) [2] and a study in Isfahan, Iran among boys and girls in middle school, 42.4% and 18.1% in high school, respectively [24] and in Uganda 59% [17]. This variation was due to differences in population characteristics and settings. The review of the above studies was done on the riskiest regions (in low and middle- income) and targets (school students). Large segments of adolescents or youth found in the school compound were much more likely to encounter violence from their peers.

In this study, the prevalence of sexual violence among youth was 3.4%. This finding was consistent with a study done in Uganda 5.6% [25] and Ethiopia 7% [12]. But lower than a review done in Nepal 29% and Cameroon 22% [16] and Uganda 35% [17]. This difference might be due to the study setting and period. Currently, the violators of sexual violence and

their legal punishment were exhibited through the media that might impede others to commit sexual violence.

The prevalence of scared/intimidated (psychological violence) in this study was 13.6%. It was higher than a study done in Serbia 2.8% [10], but this finding was lower than a study in Isfahan, Iran among boys 25% in middle school and 19.1% in high school, respectively [24] and similar study in the same country among adolescents and youth reported to be between 30% and 65.5%, respectively [26] and in Uganda 33.3% [17]. This difference could be a difference in the study area, setting, and participants; a large segment of adolescents or youth found in the school compound were much more likely to encounter violence from their peers, and the biological differences of the participants might explain some differences in levels of violence between boys and girls.

**Table 3. Factors associated with violence among youth in Northwest, Ethiopia.**

| Variable name | Violence for the past12months | | COR/CI | AOR | P-value |
|---|---|---|---|---|---|
| | No | Yes | | | |
| **Sex** | | | | | |
| Male | 601 | 179 | 1.19 (0.13, 1.51) | | |
| Female | 653 | 164 | 1 | 1 | |
| **Marital status** | | | | | |
| Single | 1010 | 261 | 1 | 1 | |
| Married | 224 | 59 | 1.02 (0.74, 1.40) | 1.77(1.13, 2.79) | 0.012* |
| Divorced | 20 | 23 | 4.45 (2.41, 8.23) | 5.67(2.93, 10.99) | 0.0001*** |
| **Educational level** | | | | | |
| Not educated | 95 | 19 | 0.59 (0.33, 1.07) | | |
| Primary education | 571 | 168 | 0.87 (0.60, 1.26) | | |
| Secondary education | 443 | 107 | 0.72 (0.49, 1.05) | | |
| Higher education | 145 | 49 | 1 | 1 | |
| **Currently attending school** | | | | | |
| No | 469 | 142 | 1.18 (0.93, 1.51) | | |
| Yes | 785 | 201 | 1 | 1 | |
| **With whom you live** | | | | | |
| Mother and father | 610 | 148 | 1 | 1 | |
| Mother only | 126 | 58 | 1.89 (1.33, 2.78) | 1.85 (1.28, 2.66) | 0.001*** |
| Father only | 28 | 17 | 2.50 (1.33, 4.69) | 2.45 (1.30, 4.63) | 0.006** |
| Husband/wife | 188 | 34 | 0.76 (0.49, 1.12) | 0.44 (0.25, 0.76) | 0.004** |
| Relative/sister/brother | 79 | 26 | 1.36 (0.84, 2.19) | 1.35 (0.83, 2.19) | 0.223 |
| Others | 61 | 9 | 0.61(0.29, 1.25) | 0.49 (0.23, 1.05) | 0.065 |
| Alone | 162 | 51 | 1.29 (0.90, 1.86) | 0.91 (0.62, 1.35) | 0.652 |
| **Wealth index** | | | | | |
| First(lowest) | 255 | 64 | 0.75 (0.52, 1.09) | | |
| Second | 255 | 65 | 0.76 (0.53, 1.10) | | |
| Third | 245 | 74 | 0.90 (0.63, 1.30) | | |
| Fourth | 260 | 60 | 0.69 (0.47, 1.01) | | |
| Highest | 239 | 80 | 1 | 1 | |
| **Substance use** | | | | | |
| **No** | 1186 | 303 | 1 | 1 | |
| **Yes** | 68 | 40 | 2.30 (1.53, 3.47) | 2.38 (1.56, 3.66) | 0.0001*** |

P value < 0.05, p value <0.01 and p value ≤ 0.001.

In the multivariate logistic regression proved that the odds of being married and divorced were 1.8 and 5.7 times more likely associated with violence among youth compared to unmarried or single, respectively. Never-married women are less likely to have experienced violence. This finding was inconsistent with a study done in Serbia [10] and DHS in Ethiopia [12]. The possible explanation for this difference could be married youth or divorced were unemployed and those who were married also economically dependent on their husband or husband's parents that led them to conflict and exposed to violence.

Living with mother only or father alone was 2 to 2.5 times more likely associated with violence compared to living with both parents. This finding was consistent with a study done in Goa, India [27] and Arbaminch town, Gammo Goffa zone, Southern Ethiopia [28]. The possible explanation could be, parental monitoring and supportiveness minimize their chance of exposure to risks of substance use and violence.

In this study substance use was 2.4 times more likely associated with youth violence compared to non-substance use. The finding of this study corroborated with studies done in different countries [21, 27, 29, 30]. This is because alcohol use directly affects cognitive and physical functioning and can reduce self-control and the ability to process information and assess risks. It can increase impulsiveness and make particular drinkers more likely to engage in violent behavior. Several studies confirm that violent incidents often occur in situations of alcohol intoxication [29, 30].

## Strength and limitations of this study

The strength of this study was community based study that could be more representative for the study area. The main limitations of this study relate to the cross-sectional design, which does not allow us to determine the direction of the causality of the detected associations. It doesn't measure the amount and frequency of substance use among the respondents. There might be recall bias and social desirability bias since the data were collected by interviewer administered questionnaire.

## Conclusions

The prevalence of violence among youth was high compared to other studies special emphasis should be given for youth violence in order to manage the victims as well as for preventing and controlling the identified factors through strengthening policies and strategies.

## Implications of the study

The prevalence of violence in this study was high. The policy makers and programme planners will give attention to prevent and control violence and immediate interventions for the victims by integrating violence prevention with other sexual and reproductive health services, community involvement and behavioral change communication for youth including intersectoral collaborations.

## Supporting information

**S1 Data.**
(SAV)

**S1 File.**
(RAR)

## Acknowledgments

The authors would like to acknowledge the University of Gondar for encouraging us to conduct this research. We have extended our gratitude for the data collectors and supervisors for their unreserved work. The authors also forwarded thanks to the study participants for providing us all the necessary information.

## Author Contributions

**Conceptualization:** Alehegn Bishaw Geremew, Abebaw Addis Gelagay, Telake Azale Bisetegn, Yohannes Ayanaw Habitu, Solomon Mekonen Abebe, Eshetie Melese Birru, Temiro Azanaw Mengistu, Yilikal Tiruneh Ayele, Hedija Yenus Yeshita.

**Data curation:** Abebaw Addis Gelagay, Telake Azale Bisetegn, Yohannes Ayanaw Habitu, Solomon Mekonen Abebe, Eshetie Melese Birru, Hedija Yenus Yeshita.

**Formal analysis:** Hedija Yenus Yeshita.

**Funding acquisition:** Abebaw Addis Gelagay.

**Investigation:** Alehegn Bishaw Geremew, Abebaw Addis Gelagay, Telake Azale Bisetegn, Yohannes Ayanaw Habitu, Solomon Mekonen Abebe, Eshetie Melese Birru, Temiro Azanaw Mengistu, Yilikal Tiruneh Ayele, Hedija Yenus Yeshita.

**Methodology:** Alehegn Bishaw Geremew, Abebaw Addis Gelagay, Telake Azale Bisetegn, Yohannes Ayanaw Habitu, Solomon Mekonen Abebe, Eshetie Melese Birru, Temiro Azanaw Mengistu, Yilikal Tiruneh Ayele, Hedija Yenus Yeshita.

**Project administration:** Alehegn Bishaw Geremew, Abebaw Addis Gelagay, Telake Azale Bisetegn, Yohannes Ayanaw Habitu, Solomon Mekonen Abebe, Eshetie Melese Birru, Temiro Azanaw Mengistu, Yilikal Tiruneh Ayele, Hedija Yenus Yeshita.

**Resources:** Hedija Yenus Yeshita.

**Software:** Abebaw Addis Gelagay, Hedija Yenus Yeshita.

**Supervision:** Alehegn Bishaw Geremew, Abebaw Addis Gelagay, Telake Azale Bisetegn, Yohannes Ayanaw Habitu, Solomon Mekonen Abebe, Eshetie Melese Birru, Temiro Azanaw Mengistu, Yilikal Tiruneh Ayele, Hedija Yenus Yeshita.

**Validation:** Alehegn Bishaw Geremew, Abebaw Addis Gelagay, Telake Azale Bisetegn, Yohannes Ayanaw Habitu, Solomon Mekonen Abebe, Eshetie Melese Birru, Temiro Azanaw Mengistu, Yilikal Tiruneh Ayele, Hedija Yenus Yeshita.

**Visualization:** Alehegn Bishaw Geremew, Abebaw Addis Gelagay, Telake Azale Bisetegn, Yohannes Ayanaw Habitu, Solomon Mekonen Abebe, Eshetie Melese Birru, Temiro Azanaw Mengistu, Yilikal Tiruneh Ayele, Hedija Yenus Yeshita.

**Writing – original draft:** Alehegn Bishaw Geremew, Abebaw Addis Gelagay.

**Writing – review & editing:** Alehegn Bishaw Geremew, Abebaw Addis Gelagay, Telake Azale Bisetegn, Yohannes Ayanaw Habitu, Solomon Mekonen Abebe, Eshetie Melese Birru, Hedija Yenus Yeshita.

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
