## [Decision Letter · Decision Letter 0]

17 Aug 2021

PONE-D-20-39684

Prevalence of violence and associated factors among Youth in Northwest Ethiopia: Community-based Cross-sectional study

PLOS ONE

Dear Dr. Yeshita,

Thank you for submitting your manuscript to PLOS ONE. After careful consideration, we feel that it has merit but does not fully meet PLOS ONE’s publication criteria as it currently stands. Therefore, we invite you to submit a revised version of the manuscript that addresses the points raised during the review process.

In particular, Reviewer 1 is not satisfied with there being no synthesis from the literature review on where the knowledge gaps are, and thus what are the relevant scientific questions addressed in this manuscript. Both reviewers are concerned whether the approach has been carefully thought through, and the methods chosen are appropriate. The authors are also asked to better organize their results and discussion to better the story behind their findings.

We look forward to receiving your revised manuscript.

Kind regards,

Siew Ann Cheong, Ph.D.

Academic Editor

PLOS ONE

Journal Requirements:

2. Please provide additional details regarding participant consent. In the ethics statement in the Methods and online submission information, please ensure that you have specified what type you obtained (for instance, written or verbal, and if verbal, how it was documented and witnessed). If your study included minors, state whether you obtained consent from parents or guardians. 

"The  University of Gondar had sponsored this study. However, it has no role in manuscript preparation and publication."

"The  University of Gondar had sponsored this study. However, it has no role in manuscript preparation and publication."

We note that one or more of the authors is affiliated with the funding organization, indicating the funder may have had some role in the design, data collection, analysis or preparation of your manuscript for publication; in other words, the funder played an indirect role through the participation of the co-authors. If the funding organization did not play a role in the study design, data collection and analysis, decision to publish, or preparation of the manuscript and only provided financial support in the form of authors' salaries and/or research materials, please do the following:

a. Review your statements relating to the author contributions, and ensure you have specifically and accurately indicated the role(s) that these authors had in your study. These amendments should be made in the online form.

b. Confirm in your cover letter that you agree with the following statement, and we will change the online submission form on your behalf: 

“The funder provided support in the form of salaries for authors [insert relevant initials], but did not have any additional role in the study design, data collection and analysis, decision to publish, or preparation of the manuscript. The specific roles of these authors are articulated in the ‘author contributions’ section.

7. PLOS requires an ORCID iD for the corresponding author in Editorial Manager on papers submitted after December 6th, 2016. Please ensure that you have an ORCID iD and that it is validated in Editorial Manager. To do this, go to ‘Update my Information’ (in the upper left-hand corner of the main menu), and click on the Fetch/Validate link next to the ORCID field. This will take you to the ORCID site and allow you to create a new iD or authenticate a pre-existing iD in Editorial Manager. Please see the following video for instructions on linking an ORCID iD to your Editorial Manager account: https://www.youtube.com/watch?v=_xcclfuvtxQ

8. Your ethics statement should only appear in the Methods section of your manuscript. If your ethics statement is written in any section besides the Methods, please move it to the Methods section and delete it from any other section. Please ensure that your ethics statement is included in your manuscript, as the ethics statement entered into the online submission form will not be published alongside your manuscript. 

Reviewers' comments:

Reviewer's Responses to Questions

**Comments to the Author**

1. Is the manuscript technically sound, and do the data support the conclusions?

Reviewer #1: No

Reviewer #2: Yes

2. Has the statistical analysis been performed appropriately and rigorously? 

Reviewer #1: No

Reviewer #2: Yes

3. Have the authors made all data underlying the findings in their manuscript fully available?

Reviewer #1: No

Reviewer #2: Yes

4. Is the manuscript presented in an intelligible fashion and written in standard English?

Reviewer #1: Yes

Reviewer #2: Yes

5. Review Comments to the Author

Reviewer #1: Thank you for the opportunity to review the manuscript titled “Prevalence of violence and associated factors among Youth in Northwest Ethiopia: Community-based Cross-sectional study”. This paper assessed the prevalence of violence and related factors among youth in Northwest Ethiopia. Overall, this manuscript is not well-written, with too broad introduction, unclear dependent and independent variables used for analysis, and unorganized results. Below are the detailed comments.

Abstract:

Page 2, line 33, the authors stated that “There was limited information on the burden of violence and factors among this age group”. This is not true. In fact, WHO has published many materials regarding youth violence, school-based violence, and prevention strategies focusing on the youth population. The CDC has also posted guidelines for youth violence prevention at individual, interpersonal, community, societal levels by reducing risk factors and promoting protective factors.

Page 2, lines 39-40, variables/risk factors used in the logistic model should be specified in the abstract. What was the dependent variable? What were the independent variables?

Page 3, lines 47-48, is it a new sentence for “Living with mother’s only... ”? This part should be checked for typo and reorganized for better understanding.

Introduction:

Page 3, line 63, need a citation for the sentence “It is estimated that each year, 200,000 homicides occur in this age group in the world.”

Pages 4-5, lines 82-108, the authors introduced many violence-related studies involving youth around the world. What was the core information the authors tried to deliver? As we all know, the prevalence of violence has been well studied; in some countries, surveillance data are available to monitor the trend and consequences of youth risk behaviors including violence. The authors may want to reorganize the literature and see how they can better serve their paper.

Page 6, lines 123-129, what are the gaps in the current literature? The authors need to briefly summarize the findings from pervious violence-related studies that included youth in Ethiopia, and then specify the gaps that this study may bridge. If very limited studies were found, studies in other countries with similar demographic characteristics or socio-economic status should be highlighted to provided information instead.

Methods:

Page 7, lines 147-149, the authors mentioned: “Firstly, two woredas from the central Gondar zone, one woreda from each North Gondar zone and west Gondar zone were selected. Secondly, three kebeles were selected from the selected woredas.” How were the woredas and kebeles selected? Details need to be provided.

Pages 7-8, lines 154-167, what were the variables/survey questions used in this study? Only definitions from WHO are insufficient for readers to learn this study. The authors may benefit from providing a detailed description of the survey, including how it was developed, included domains/topics, and questions/valid responses used in this study.

Pages 8-9, lines 180-185, was it a complete dataset for analysis? Any missing data information?

Results:

Pages 9-13, table 1 and table 2 can be combined, as both addressed the participants’ demographic information.

Page 13, lines 201-205, it seems there were multiple violence variables. Which one was the dependent variable used for the logistic regression?

Page 15, line 201, what were the independent variables that tested in the regression? If all of them were listed in table 4, why they were selected? Are there any other variables in the survey that can be potentially taken into the model (e.g., mental health problems)?

Discussion:

This section repeated too much of the results (e.g., lines 227, 235, 248, 251...). Also, the authors may want to discuss the strengths of this study and implications for violence prevention.

Reviewer #2: This research article has chosen well objectives, study selection, study characteristic however there is limited information about assessing risk of bias. Authors have IRB authorizations and followed the scientific research.

6. PLOS authors have the option to publish the peer review history of their article (what does this mean?). If published, this will include your full peer review and any attached files.

Reviewer #1: No

Reviewer #2: No

---

## [Author Response · Author response to Decision Letter 0]

16 Oct 2021

To: Dear, Siew Ann Cheong, Ph.D

PLOS ONE 

Subject: Submitting a revised version of manuscript and point by point response

PONE-D-20-39684

Prevalence of violence and associated factors among Youth in Northwest Ethiopia: Community-based Cross-sectional study

Point by point response

Response for Editor

Please ensure that your manuscript meets PLOS ONE's style requirements, Thank You Dear, 

• Authors have made corrections on the manuscript by using the PLOS ONE’s style.

2. Please provide additional details regarding participant consent. In the ethics statement in the Methods and online submission information. • This is also corrected in the manuscript

3. Please update your submission to use the PLOS LaTeX template. • Authors, We have done correction based on this instruction

4. The University of Gondar had sponsored this study. However, it has no role in manuscript preparation and publication." At this time, please address the following queries:

 a) The University of Gondar supported the staffs financially to conduct this research.

b) The funder had no role in this research: had no role in study design, data collection and analysis, decision to publish, or preparation of the manuscript. 

c) All authors are received their salaries from University of Gondar except Temiro Azanaw Mengistu and Yilikal Tiruneh Ayele. 

5. Thank you for stating the following: 

a. Review your statements relating to the author contributions, and ensure you have specifically and accurately indicated the role(s) that these authors had in your study. These amendments should be made in the online form. 

b. Confirm in your cover letter that you agree with the following statement, and we will change the online submission form on your behalf: 

“The funder provided support in the form of salaries for authors [insert relevant initials], but did not have any additional role in the study design, data collection and analysis, decision to publish, or preparation of the manuscript. The specific roles of these authors are articulated in the ‘author contributions’ section. • "The University of Gondar had sponsored this study. However, it has no role in manuscript preparation and publication."

a) Authors, we have made correction in the main document. 

b) Corrected.

6. In your Data Availability statement, you have not specified where the minimal data set underlying the results described in your manuscript can be found • The minimal data set underlying the results are found in the tables within the manuscript.

7. PLOS requires an ORCID iD for the corresponding author in Editorial Manager on papers submitted after December 6th, 2016. Please ensure that you have an ORCID ID and that it is validated in Editorial Manager Thank You.

8. Your ethics statement should only appear in the Methods section of your manuscript. If your ethics statement is written in any section besides the Methods, please move it to the Methods section and delete it from any other section • It is moved to the appropriate place.

Reviewers' comments

Reviewer #1: 

Abstract:

Page 2, line 33, the authors stated that “There was limited information on the burden of violence and factors among this age group”. This is not true • Thank you for providing an important comments and questions. 

• Even if there were a lot of studies in different setting, those studies mainly concerned on institution based, the study participants were adolescent females or males only, researched a single problem (sexual or physical). Whereas, this study was community based including both sex group, assessed overall prevalence of violence in general and the three common forms of violence in particular. 

Page 2, lines 39-40, variables/risk factors used in the logistic model should be specified in the abstract. What was the dependent variable? What were the independent variables?

 Thank You! 

• Youth violence was the dependent variable whereas socio-demographic variables(age, sex, marital status, religion, residence, occupation, educational level, currently attending school, with whom you live, wealth index) and substance use were the independent variables.

Page 3, lines 47-48, is it a new sentence for “Living with mother’s only... ”? This part should be checked for typo and reorganized for better understanding. • It has no any error. When the data collector asked an individual’s living arrangement. The response of the participant was living with mother’s only meaning the respondent was living only with his or her mother. 

Introduction:

Page 3, line 63, need a citation for the sentence “It is estimated that each year, 200,000 homicides occur in this age group in the world.”. Thank you, 

• Authors accepted the comments and corrected it accordingly.

Pages 4-5, lines 82-108, the authors introduced many violence-related studies involving youth around the world. What was the core information the authors tried to deliver? As we all know, the prevalence of violence has been well studied; in some countries, surveillance data are available to monitor the trend and consequences of youth risk behaviors including violence. The authors may want to reorganize the literature and see how they can better serve their paper. • The core information was to know how much the magnitude of youth violence in different countries and settings in general and types of violence in particular and also to look-over a limited studies on youth violence in Ethiopia. These reviews used for the authors to compare their own research findings accordingly. 

Page 6, lines 123-129, what are the gaps in the current literature? The authors need to briefly summarize the findings from pervious violence-related studies that included youth in Ethiopia, and then specify the gaps that this study may bridge. If very limited studies were found, studies in other countries with similar demographic characteristics or socio-economic status should be highlighted to provided information instead.

 • The gaps in these limited literatures were the studies mainly concentrated among women age 15 to 49 or only on males, and institutional based. Whereas, this study concerned on both sex, among youth and community based and also assessed the three forms of violence simultaneously. 

• Studies in other countries with similar demographic characteristics or socio-economic status might have different socio-cultural and behavioral characteristics. There might be also a variation on the associated factors. So, Identifying the prevalence and associated factors of youth violence was helpful to design an intervention in the study area. 

Methods:

Page 7, lines 147-149, the authors mentioned: “Firstly, two woredas from the central Gondar zone, one woreda from each North Gondar zone and west Gondar zone were selected. Secondly, three kebeles were selected from the selected woredas.” How were the woredas and kebeles selected? Details need to be provided. • Centeral Gondar having 13 woredas (north Gondar zone having 7 woredas and west Gondar zone having 4 woredas). 

“Firstly, two woredas from the central Gondar zone, one woreda from each North Gondar zone and west Gondar zone were selected by using simple random sampling tecnique. 

Secondly, from the total number of kebeles (546 kebeles) in the three zones: three kebeles were selected from each selected woredas by simple randomly sampling.

Pages 7-8, lines 154-167, what were the variables/survey questions used in this study? Only definitions from WHO are insufficient for readers to learn this study. The authors may benefit from providing a detailed description of the survey, including how it was developed, included domains/topics, and questions/valid responses used in this study.

 The survey questions were

• Have you experienced violence in the past 12 months?

• Have you experienced violence like hit /slapped or thrown something by someone in the past 12 months?

• Have you experienced violence like scared/intimidated by somebody in the past 12 months?

• Have you exposed to sexual violence for the last 12 months?

Pages 8-9, lines 180-185, was it a complete dataset for analysis? Any missing data information? • Yes, No any missing data information

Results:

Pages 9-13, table 1 and table 2 can be combined, as both addressed the participants’ demographic information Thank You, 

• Table 1 and table 2 are merged into one table.

Page 13, lines 201-205, it seems there were multiple violence variables. Which one was the dependent variable used for the logistic regression?

 • The dependent variable used for the logistic regression was “prevalence of violence” 

Page 15, line 201, what were the independent variables that tested in the regression? If all of them were listed in table 4, why they were selected? Are there any other variables in the survey that can be potentially taken into the model (e.g., mental health problems)? • No, all variables listed in the socio-demographic table were tested in the bi-variable analysis. Those variables have a p-value of < 0.2 were taken in to the multivariable logistic regression model. Those variables are listed in the table 4(3) and other potential variable like substance use was considered from the survey.

Discussion:

This section repeated too much of the results (e.g., lines 227, 235, 248, 251...). Also, the authors may want to discuss the strengths of this study and implications for violence prevention • It has no any repetition; the authors wanted to describe how much the prevalence of different forms of youth violence (physical, sexual or emotional) and compared them with other studies. 

Reviewer #2

There is limited information about assessing risk of bias Thank you very much!

• In addition to recall bias, social desirability bias might be occurred during data collection since the data collectors were used interviewer administered questionnaires.

---

## [Decision Letter · Decision Letter 1]

25 Nov 2021

PONE-D-20-39684R1Prevalence of violence and associated factors among Youth in Northwest Ethiopia:

Community-based Cross-sectional studyPLOS ONE

Dear Dr. Yeshita,

Thank you for submitting your manuscript to PLOS ONE. After careful consideration, we feel that it has merit but does not fully meet PLOS ONE’s publication criteria as it currently stands. Therefore, we invite you to submit a revised version of the manuscript that addresses the points raised during the review process.

In particular, Reviewer 1 complained that most of his comments were not addressed in the first revision. Please be sure to address all these comments.

We look forward to receiving your revised manuscript.

Kind regards,

Siew Ann Cheong, Ph.D.

Academic Editor

PLOS ONE

Reviewers' comments:

Reviewer's Responses to Questions

**Comments to the Author**

1. If the authors have adequately addressed your comments raised in a previous round of review and you feel that this manuscript is now acceptable for publication, you may indicate that here to bypass the “Comments to the Author” section, enter your conflict of interest statement in the “Confidential to Editor” section, and submit your "Accept" recommendation.

Reviewer #1: (No Response)

Reviewer #2: All comments have been addressed

2. Is the manuscript technically sound, and do the data support the conclusions?

Reviewer #1: Partly

Reviewer #2: Yes

3. Has the statistical analysis been performed appropriately and rigorously? 

Reviewer #1: (No Response)

Reviewer #2: I Don't Know

4. Have the authors made all data underlying the findings in their manuscript fully available?

Reviewer #1: (No Response)

Reviewer #2: Yes

5. Is the manuscript presented in an intelligible fashion and written in standard English?

Reviewer #1: No

Reviewer #2: Yes

6. Review Comments to the Author

Reviewer #1: Thank you for the opportunity to review the revised manuscript titled “Prevalence of violence and associated factors among Youth in Northwest Ethiopia: Community-based Cross-sectional study”. Without using track changes in the main text or pointing out the changes in the responses to comments, it was very challenging to tell the difference between the original submission and the revised one. I did a side-by-side check and found there were very limited changes in the introduction, methods, results, and discussion sections. Most of my previous concerns/suggestions remained unsolved.

Reviewer #2: The author/authors have responded to all the comments and provided information regarding their research .

7. PLOS authors have the option to publish the peer review history of their article (what does this mean?). If published, this will include your full peer review and any attached files.

Reviewer #1: No

Reviewer #2: **Yes: **Ashraf Mozayani

---

## [Author Response · Author response to Decision Letter 1]

5 Jan 2022

Point by point response

Response for Editor 

Please ensure that your manuscript meets PLOS ONE's style requirements, Dear Editor, we have tried to address the PLOS ONE’s requirements.

Reviewer #1: 

Abstract:

Page 2, line 33, the authors stated that “There was limited information on the burden of violence and factors among this age group”. This is not true. Dear reviewer #1, Author’s have accepted your complaint, however, we couldn’t understand which areas/questions were not well addressed.

Anyway, we have made modification on some questions as much as possible. 

• Authors reviewed a lot of studies however, those studies mainly concerned on institution based, the study participants were adolescent females or males only, researched a single problem (sexual or physical) and didn’t show burden of violence among this specific age group. Whereas, this study was community based including both sex group, assessed overall prevalence of violence in general and the three common forms of violence in particular. 

Page 2, lines 39-40, variables/risk factors used in the logistic model should be specified in the abstract. What was the dependent variable? What were the independent variables? Thank You! 

• “Youth violence” was the dependent variable whereas all socio-demographic variables listed in table1 and substance use were the independent variables.

Page 3, lines 47-48, is it a new sentence for “Living with mother’s only... ”? This part should be checked for typo and reorganized for better understanding. • Living with mother’s only indicates adolescents/youth were living only with the support of his or her mother. When the data collectors asked them about currently living arrangement of the participants, a numbers of the participant’s responses were living with mother only or father only or living with both mother and father or others options found in the table. 

Introduction:

Page 3, line 63, need a citation for the sentence “It is estimated that each year, 200,000 homicides occur in this age group in the world.”. Thank you, 

• Authors accepted the comment and corrected it accordingly.

Pages 4-5, lines 82-108, the authors introduced many violence-related studies involving youth around the world. What was the core information the authors tried to deliver? As we all know, the prevalence of violence has been well studied; in some countries, surveillance data are available to monitor the trend and consequences of youth risk behaviors including violence. The authors may want to reorganize the literature and see how they can better serve their paper. • The core information that the authors tried to deliver was how much the prevalence of youth violence in different countries and settings in general and types of violence in particular. By reviewing different literature, authors identified limited studies on youth violence in Ethiopia, most of the studies were concerned on gender based or intimate partner violence among reproductive age women from 15 to 49 or sexual and reproductive health problems among youth.

Page 6, lines 123-129, what are the gaps in the current literature? The authors need to briefly summarize the findings from pervious violence-related studies that included youth in Ethiopia, and then specify the gaps that this study may bridge. If very limited studies were found, studies in other countries with similar demographic characteristics or socio-economic status should be highlighted to provided information instead.

 • The gaps in these limited literatures were the studies mainly concentrated among women age 15 to 49 years. On the other hand, analysis from EDHS data that might underestimate the prevalence of violence, other studies were done only on males on a single type of violence or institutional based. Whereas, this study concerned youth violence on both sex groups, community based and also assessed the three forms of violence among youth simultaneously. 

• Studies in other countries with similar demographic characteristics or socio-economic status might have different socio-cultural and behavioral characteristics. There might be also a variation on the associated factors due to that their results might not represent the results of studies in our country. Therefore, Identifying the prevalence and associated factors of youth violence will be helpful to design prevention and intervention strategies in the study area. 

Methods:

Page 7, lines 147-149, the authors mentioned: “Firstly, two woredas from the central Gondar zone, one woreda from each North Gondar zone and west Gondar zone were selected. Secondly, three kebeles were selected from the selected woredas.” How were the woredas and kebeles selected? Details need to be provided. • This mega project study was done in centeral, west and north Gondar zone. A total 24 woredas present in the three zone: centaral Gondar zone have 13 woredas, north Gondar zone 7 woredas , and west Gondar zone 4 woredas. 

“Firstly, two woredas from the central Gondar zone, one woreda from each north Gondar zone and west Gondar zone were selected by using simple random sampling technique. 

Secondly, from the total number of kebeles (546 kebeles) in the three zones: three kebeles were selected from each selected woredas by simple randomly sampling.

Pages 7-8, lines 154-167, what were the variables/survey questions used in this study? Only definitions from WHO are insufficient for readers to learn this study. The authors may benefit from providing a detailed description of the survey, including how it was developed, included domains/topics, and questions/valid responses used in this study.

 Thank You, we have addressed the survey questions in the document.

The survey questions were

• Have you experienced violence in the past 12 months?

• Have you experienced violence like hit /slapped or thrown something by someone in the past 12 months?

• Have you experienced violence like scared/intimidated by somebody in the past 12 months?

• Have you exposed to sexual violence for the last 12 months?

Pages 8-9, lines 180-185, was it a complete dataset for analysis? Any missing data information? • Yes 

• No any missing data information was found in this study analysis.

Results:

Pages 9-13, table 1 and table 2 can be combined, as both addressed the participants’ demographic information Thank You, 

• We have done it.

Page 13, lines 201-205, it seems there were multiple violence variables. Which one was the dependent variable used for the logistic regression?

 • The dependent variable used for the logistic regression was “prevalence of youth violence” 

Page 15, line 201, what were the independent variables that tested in the regression? If all of them were listed in table 4, why they were selected? Are there any other variables in the survey that can be potentially taken into the model (e.g., mental health problems)? • All independent variables listed in the socio-demographic table and other potential variable from the survey (substance use) were tested in the bi-variable analysis. 

• Variables listed in table 4(3) (sex, marital status, educational level, currently attending with school, living arrangement, wealth index, and substance use) were selected after bivariable analysis results- variables have a p-value of < 0.2 were taken in to the multivariable logistic regression model.

• Potential variable selected in the survey was substance use. 

Discussion:

This section repeated too much of the results (e.g., lines 227, 235, 248, 251...). Also, the authors may want to discuss the strengths of this study and implications for violence prevention. • Thank you. Authors have made correction on the main document.

• The strength and implication of this study is included in the main document.

---

## [Decision Letter · Decision Letter 2]

16 Feb 2022

Prevalence of violence and associated factors among Youth in Northwest Ethiopia:

Community-based Cross-sectional study

PONE-D-20-39684R2

Dear Dr. Yeshita,

We’re pleased to inform you that your manuscript has been judged scientifically suitable for publication and will be formally accepted for publication once it meets all outstanding technical requirements.

Kind regards,

Siew Ann Cheong, Ph.D.

Academic Editor

PLOS ONE

Additional Editor Comments (optional):

Reviewers' comments:

Reviewer's Responses to Questions

**Comments to the Author**

1. If the authors have adequately addressed your comments raised in a previous round of review and you feel that this manuscript is now acceptable for publication, you may indicate that here to bypass the “Comments to the Author” section, enter your conflict of interest statement in the “Confidential to Editor” section, and submit your "Accept" recommendation.

Reviewer #3: All comments have been addressed

2. Is the manuscript technically sound, and do the data support the conclusions?

Reviewer #3: Yes

3. Has the statistical analysis been performed appropriately and rigorously? 

Reviewer #3: Yes

4. Have the authors made all data underlying the findings in their manuscript fully available?

Reviewer #3: Yes

5. Is the manuscript presented in an intelligible fashion and written in standard English?

Reviewer #3: Yes

6. Review Comments to the Author

Reviewer #3: Very thorough feedback was give to the authors by each reviewers. Given the time and line by line response by the authors, there have been an excellent attempt to edit the manuscript accordingly. The responses are satisfactory and there there are no concerns or worry about methodology, writing, or findings at this time. Thank you for your efforts

7. PLOS authors have the option to publish the peer review history of their article (what does this mean?). If published, this will include your full peer review and any attached files.

Reviewer #3: **Yes: **Carolyn Gentle-Genitty, PhD

---

## [Editor Report · Acceptance letter]

4 Mar 2022

PONE-D-20-39684R2 

Prevalence of violence and associated factors among Youth in Northwest Ethiopia:
Community-based Cross-sectional study 

Dear Dr. Yeshita:

I'm pleased to inform you that your manuscript has been deemed suitable for publication in PLOS ONE. Congratulations! Your manuscript is now with our production department. 

Kind regards, 

on behalf of

Dr. Siew Ann Cheong 

Academic Editor

PLOS ONE